# Differences in Mechanical, Electrical and Calcium Transient Performance of the Isolated Right Atrial and Ventricular Myocardium of Guinea Pigs at Different Preloads (Lengths)

**DOI:** 10.3390/ijms242115524

**Published:** 2023-10-24

**Authors:** Ruslan Lisin, Alexandr Balakin, Elena Mukhlynina, Yuri Protsenko

**Affiliations:** Institute of Immunology and Physiology, Ural Branch of Russian Academy of Sciences, 106 Pervomayskaya Str., Yekaterinburg 620049, Russia; lisin.ruslan@gmail.com (R.L.); elena.mukhlynina@yandex.ru (E.M.); y.protsenko@iip.uran.ru (Y.P.)

**Keywords:** guinea pig myocardium, right atrium, right ventricle, Frank–Starling mechanism, Ca^2+^ transient, transmembrane action potential

## Abstract

There are only a few studies devoted to the comparative and simultaneous study of the mechanisms of the length-dependent regulation of atrial and ventricular contractility. Therefore, an isometric force-length protocol was applied to isolated guinea pig right atrial (RA) strips and ventricular (RV) trabeculae, with a simultaneous measurement of force (Frank–Starling mechanism) and Ca^2+^ transients (CaT) or transmembrane action potentials (AP). Over the entire length-range studied, the duration of isometric contraction, CaT and AP, were shorter in the RA myocardium than in the RV myocardium. The RA myocardium was stiffer than the RV myocardium. With the increasing length of the RA and RV myocardium, the amplitude and duration of isometric contraction and CaT increased, as well as the amplitude and area of the “CaT difference curves” (shown for the first time). However, the rates of the tension development and relaxation decreased. No contribution of AP duration to the heterometric regulation of isometric tension was found in either the RA or RV myocardium of the guinea pig. Changes in the degree of overlap of the contractile proteins of the guinea pig RA and RV myocardium mainly affect CaT kinetics but not AP duration.

## 1. Introduction

The Frank–Starling mechanisms for the heterometric autoregulation of contractility of the atrial and ventricular cardiomyocytes play an important role in the regulation of pump function and the mutual adjustment of the chambers of a healthy and pathologically altered heart [1]. The structural and functional characteristics of the atrial and ventricular myocardium are mutually coordinated for effective interaction during the cardiac cycle [2] while maintaining the “correct activation pattern” [3]. It has long and repeatedly been demonstrated that atrial systole has a powerful augmentative effect on ejection volume in a healthy heart. Its effect changes with the development of cardiac pathology [3,4,5,6,7].

In all animals and in humans, there are considerable morphofunctional differences between the atrial and ventricular myocardium. The atrial myocardium contains slightly fewer cardiomyocytes and more fibroblasts per unit volume than the ventricular myocardium. Both the volume and the cross-sectional area of the atrial cells are smaller than those of the ventricular cells in the guinea pig and other animals [8]. Atrial cardiomyocytes have a different ultrastructure than ventricular myocytes. They also have different gene expression patterns, e.g., for structural protein transcription factors and ion channels [9]. Markedly different transcriptional signatures have been found in ventricular and atrial cardiomyocytes, reflecting developmental origins and differences in electrophysiological, contractile and secretory processes [10]. In guinea pig atrial cardiomyocytes, the relative population of sarcoplasmic reticulum (SR), in the form of two different types of peripheral junctional sarcoplasmic reticulum, is slightly higher and also has significantly fewer t-tubules than in ventricular cardiomyocytes [11]. The morphological features are reflected in the characteristics of all atrial and ventricular myocardial functions.

Atrial and ventricular contractions have different mechanical characteristics, including the rate of contraction, the volume of blood supplied and the range of pressure generated. The human ventricle predominantly expresses β-cardiac myosin, whereas the atrium predominantly expresses the α-isoform. The two myosin isoforms, α and β, although 91% identical in the motor domain sequence [12], have different mechanical and biochemical properties. The mechanical stress produced by the ventricular myocardium is greater (by approximately 30–50%) and the duration of the entire contraction–relaxation cycle is almost twice as long in the ventricle as in the atrium [13].

Although the electrophysiological properties of atrial and ventricular myocytes are largely similar, they differ in several features so as to meet the functional requirements of the different chambers of the heart [14]. Part of the difference in the shape of transmembrane action potential (AP) signals from atrial and ventricular myocytes is due to differences in the expression of potassium potential-dependent channels [15,16,17]. When rat atrial and ventricular RNA samples were compared, transcripts of the Kvl.2 and Kv4.2 genes showed the greatest differences in relative abundance [18]. Heterogeneity of cell membrane sodium current (*I*_Na_) was found in atrial and ventricular myocytes of guinea pig hearts. The density of inwardly directed *I*_Na_ is approximately 50% greater in atrial myocytes than in ventricular myocytes. The time constants of *I*_Na_ activation (τ_m_) and inactivation (τ_h_) are twice as fast in atrial myocytes as in ventricular myocytes. The recovery of *I*_Na_ after inactivation is slower in atrial myocytes than in ventricular myocytes, whereas the development of inactivation at rest is faster in atrial than in ventricular myocytes [19].

In both atrial and ventricular cardiomyocytes, most of the calcium ions (Ca^2+^) required for contraction are released from the SR in response to small amounts of Ca^2+^ entering the cell through *L*-type calcium channels during the AP [20]. The recirculating Ca^2+^ fraction in atrial muscles is almost two times higher than that in guinea pig ventricular muscles [21]. Since the AP modulates the calcium transient (CaT) [22] and vice versa [23], changes in intracellular calcium concentration ([Ca^2+^]_i_) during stretching should influence the AP shape of the myocyte. In the human, large mammalian and rodent myocardium, atria of all species have been shown to have a more rapid time course of increase and decrease in [Ca^2+^]_i_ [24,25,26,27,28,29]. The reasons for the faster calcium turnover in atrial myocytes are differences in the expression and function of proteins that couple excitation and contraction, as well as differences in the ultrastructural features of the sarcolemmal and sarcoplasmic reticulum, including the extent of transverse tubules and the abundance of noncontacting SRs [30,31].

Currently, studies on contractile proteins and the contractility of both the atrial and ventricular myocardium simultaneously are beginning to appear, especially in humans [13,32]. Despite the proven importance of studying atrioventricular coordination for effective cardiac pump function, and the established individual differences in morphofunctional characteristics, we could not find any work on the comparative study of length-dependent mechanisms of regulation of atrial and ventricular contractility. In light of morphological differences in the structure of the sarcoplasmic reticulum, a comparison of the temporal and spatial characteristics of CaTs in atrial and ventricular cardiomyocytes may provide clues as to how cellular organisation affects the behaviour of the electromechanical coupling mechanism at different preloads [33]. With regard to the present work, it should be emphasised that there is a lack of data studying the effects of sequential changes in the myocardial length of the different heart chambers of different animals, including the guinea pig, on electromechanical coupling, with the exception of our work on the rat heart [34]. The present study was performed on guinea pig myocardial preparations because their myocardium differs from the rat myocardium in electromechanical coupling and is similar to the human myocardium [21,35,36].

The aim is to establish the effects of length (preload) on the basic mechanisms of the heterometric regulation of the atrial and ventricular myocardial contractility of the same heart.

Objective: to compare the characteristics of electro-mechano-calcium components of the mechanism of length-dependent regulation of contractility of isolated guinea pig right atrial and ventricular myocardial strips.

## 2. Results

### 2.1. Morphology of the Free Wall of the Right Ventricle and the Wall of the Right Atrium

The epicardium of the intact guinea pig right ventricle (RV) consists of a thin lamina of connective tissue covered with mesothelium on the free surface. The mesothelial layer of the epicardial lamina is preserved throughout. The epicardium contains various cellular elements of connective tissue, including neutrophils and lymphocytes (Figure 1A,C).

The myocardium of the RV is represented by multidirectional bundles of cardiomyocytes that are surrounded with fibres of connective tissue (Figure 1). The nuclei of the cardiomyocytes show a mild polymorphism (Figure 1A). Between the muscle elements of the myocardium are thin layers of loose connective tissue, clearly visible in sections stained with picrosirius red (Figure 1C), as well as numerous blood and lymph capillaries. The endocardium of the RV covers the interior of the heart and is visually represented by endothelium and connective tissue fibres under physiological conditions (not shown).

The epicardium of the right atrium (RA) of the guinea pig visually has a thicker connective tissue layer than that of the RV (Figure 1B,D), which is also confirmed by the data on the percentage of collagen content (Table 1). The cardiomyocytes of the RA of intact guinea pigs are also arranged in bundles, but they have a smaller diameter (Table 1) than the RV cardiomyocytes (Figure 1B,D). The nuclei of both RA and RV cardiomyocytes show a mild polymorphism (Figure 1A,B). The RA contains more nuclei per unit area than the right ventricle, but the differences are not significant (Table 1). In the atrial myocardium, interlayers of loose connective tissue between the cardiomyocytes are also clearly visible in sections stained with picrosirius red (Figure 1D). The auricles of the RA myocardium consist of multidirectional bundles of cardiomyocytes that do not form densely packed layers as in the RV myocardium. Connective tissue cells, lymphocytes and neutrophils are found in the connective tissue layers of both the atria and ventricles, but do not form significant clusters (Figure 1).

The morphometric studies show that the thickness of the RV free wall in guinea pigs is significantly greater than that of the RA wall under physiological conditions. In addition, the RV cardiomyocytes have a significantly larger diameter than the RA cardiomyocytes (Table 1).

### 2.2. Effect of Length (Preload) on Force Development

The typical behaviour of isometric force curves of guinea pig right atrial and ventricular myocardial strips in relation to the degree of length stretch or preload (an analogue of the Frank–Starling phenomenon) is shown in (Figure 2).

The values of active isometric tension amplitudes at different lengths show greater variability in the right ventricular trabeculae compared to the atrial strips. Statistically significant differences in the active isometric tension amplitudes between the RA and RV are present only at the maximal length—*L_max_* (Figure 3A). The passive tension values of the RA strips are statistically significantly higher than those of the RV trabeculae over the entire length-range studied (Figure 3B). However, all RA strips are more pliable and can be stretched more in relation to their initial length than RV trabeculae (Figure 3).

The time to peak isometric tension (TTP) and the time of tension relaxation from TTP to half the amplitude of active tension in the relaxation phase (T50) are significantly shorter in RA strips compared to those in RV trabeculae over the entire length range studied (*p* < 0.001, Mann–Whitney U-test) (Figure 4A,B). An increase in the length of the RA strips and RV trabeculae leads to an increase in TTP and T50 values (Figure 4A,B). Changes in the length of the muscle preparations have a statistically significant effect on TTP and T50, both in the RA strips and in the RV trabeculae (*p* < 0.05, Friedman ANOVA). The detailed results of the statistical tests can be found in the Appendix A. The values of the maximal rates of isometric tension development ((dP/dt)/P_0 max_ development) and relaxation ((dP/dt)/P_0 max_ relaxation), normalised by the value of the force amplitude, decrease with the increasing length of both atrial strips and ventricular trabeculae (Figure 4C,D). Changes in the length of the muscle preparations have a statistically significant effect on (dP/dt)/P_0 max_ development and (dP/dt)/P_0 max_ relaxation, both in the RA strips and in the RV trabeculae (*p* < 0.05, Friedman ANOVA). The detailed results of the statistical tests can be found in the Appendix A. In contrast to the temporal characteristics, the differences in the normalised values of the maximal rates of tension development and relaxation between RA and RV preparations are significant not for the entire length range (*p* < 0.05, Mann–Whitney U-test). The differences are more pronounced in the values for maximal rates of tension development (Figure 4C,D).

It seems important to indirectly compare the effect of the degree of contractile protein overlap on the intracellular calcium ion release and sequestration systems in RA and RV myocardial preparations. For this purpose, it is necessary to determine the correlation coefficients between the normalised values of maximal rates of force development and relaxation over the entire length-range studied (Table 2). High values of the Spearman correlation coefficients show a strong direct linear relationship between the values of maximal rates of force development and a relaxation of the isometric tension of the RA strips (statistically insignificant). These parameters should correlate if length changes have the same effects on the systems of calcium ion release and sequestration. However, the RV trabeculae show only a moderate linear relationship between these parameters over the entire length range studied (statistically insignificant).

### 2.3. Effect of Length (Preload) on the Action Potential

The transmembrane action potential duration (APD) of the cardiomyocytes of the guinea pig right atrial myocardial multicellular strips during the repolarisation phase at the level of amplitude decay by 90% (APD90) is shorter than the APD90 of the right ventricular cardiomyocytes over the entire length range studied (*p* < 0.05, Mann–Whitney U-test) (Figure 5B). There is no effect of the length of the guinea pig right atrial (*n* = 12) and ventricular (*n* = 13) myocardium on the APD90 (*p* > 0.05, Friedman ANOVA) (Figure 5B). The detailed results of the statistical tests can be found in the Appendix A. The duration of the APD90 in guinea pig atrial preparations at a length of 0.95 *L_max_* is 81 ± 6 msec and, in ventricular preparations, 172 ± 12 msec.

### 2.4. Effect of Length (Preload) on the Calcium Transient

An increase in the length of the RA strips and RV trabeculae leads to a decrease in the duration of the CaT at the beginning of decay and an increase in duration at the end of decay—the “crossover” phenomenon (Figure 6A,C). Changes in the length of the preparations have a statistically significant effect on the time interval from the moment of the peak of the CaT to the level of the amplitude decline by 70% (T70Ca), both in the RA strips and in the RV trabeculae (*p* < 0.05, Friedman ANOVA) (Figure 6B). The detailed results of the statistical tests can be found in the Appendix A. It is worth noting that the post hoc test for all groups did not reveal significant differences for T70Ca at different muscle preparation lengths in the RV group.

Representative examples of the “CaT difference curves” for the RA and RV muscle strips are shown in (Figure 7A,B). They are the result of subtraction of the CaT trajectory values obtained at a length of 0.8 *L_max_* from the CaT trajectories obtained at different lengths of the atrial strips or ventricular trabeculae (see Section 4 Materials and methods). Using the amplitude and area of the shaded segment of the difference curve (the III phase of “the CaT difference curve”), we can indirectly assess the effects of the degree of contractile proteins overlap on the release (uncoupling from troponin C (TnC)) and sequestration of calcium ions from the myoplasm of cardiomyocytes (Figure 7B). It is shown for the first time that the amplitude and area of the “CaT difference curves” increase with the increasing length of the guinea pig RA strips and RV trabeculae (*p* < 0.05, Friedman ANOVA). The detailed results of the statistical tests can be found in the Appendix A. The differences in the values of the amplitudes and relative areas (in %) of the III phase of the “CaT difference curve” between the RA strips and RV trabeculae in the length range 0.8–1.0 *L_max_* (*p* < 0.05, Mann–Whitney U-test) are insignificant (Figure 7C,D).

## 3. Discussion

### 3.1. Active and Passive Components of Isometric Tension

The amplitude of active, and the magnitude of passive, isometric tension of isolated guinea pig right atrial myocardial strips and ventricular trabeculae increase monotonically with the increasing degree of stretch of the muscle preparations in our experiments (Figure 3). However, statistically significant differences in the active tension amplitudes between the RA and RV myocardial preparations are only observed for the maximal length—*L_max_*. The RV trabeculae show a greater scatter in active tension values compared to those of the RA strips (Figure 3A). This phenomenon can be explained by the heterogeneity of the shapes of the RV trabeculae. In each heart, the trabeculae are arranged differently and have different lengths, widths and thicknesses. In the atria, on the other hand, it is possible to cut a strip of tissue with the same shape in the same location, thus reducing the inhomogeneity of the strip shape. It is worth noting that, in contrast to guinea pig muscle preparations, the active tension of rat RV trabeculae is significantly greater than the active tension of rat RA strips [34]. In our experiments, a higher level of passive tension was observed for atrial strips compared to ventricular trabeculae over the entire length-range studied. There are few data in the literature comparing the magnitude of atrial and ventricular passive tension in a single heart. It is known that softer N2BA titin is expressed in the left atrium of cattle in vitro [37,38], and greater ventricular stiffness compared with atrial stiffness has been observed in humans in vivo [39]. At first glance, our results seem to contradict the literature. However, in our experiments, the RV trabeculae do not contain epicardium, whereas the RA strips contain epicardium with its abundance of collagen fibres (Figure 1). Furthermore, histological analysis revealed a significantly higher collagen content in the myocardium of the right atrium than in the ventricle (Table 1). Thus, the higher passive tension in the RA strips can be explained by the presence of a large amount of connective tissue in these strips.

### 3.2. Correlation Coefficients of the Maximal Rates of Tension Development and Relaxation

A stronger linear relationship between the values of the maximal rate of tension development and the maximal rate of relaxation for RA myocardial strips compared with RV trabeculae (Table 2) suggests the presence of mechanisms that are similarly modulated by stretch in the RA but not in the RV. This could be due to faster Ca^2+^ kinetics [11] and faster kinetics of atrial cross-bridges compared to those of the ventricle [12,40]. Note that such a strong linear relationship between the rates of tension-development and relaxation in the RA ensures a rapid relaxation of the atrium before the onset of ventricular systole in the whole heart. It can be assumed that the presence of such a relationship is necessary to coordinate the functioning of the heart chambers.

### 3.3. Transmembrane Action Potential

Regarding the differences in transmembrane potentials between the atrium and ventricle, our results show that the APD90 of atrial cardiomyocytes is significantly lower than the APD90 of ventricular cardiomyocytes (Figure 5B). At a length of 0.95 *L_max_*, their values are 82.3 ± 7.7 ms and 168.4 ± 15 ms, respectively (mean ± SD). A similar ratio of the duration of APs in the cells of the atria and ventricles of a guinea pig (atrium = 141 ms, ventricle = 497 ms) was shown in [41]. The authors explained their result by different properties of background K+ channels with the same conductance but slower kinetics in ventricular myocytes.

It is known that heart muscle stretch can alter the electrophysiological properties of the heart, a phenomenon known as “mechano-electrical” feedback [42,43,44,45]. However, there are contradictory data on the effects of stretch on AP duration, which can be classified as follows. Both a decrease in APD [44,46,47,48,49,50] and an increase in APD [44,51,52,53] after an imposed stretch in ventricular cardiomyocytes of guinea pigs have been shown. In addition, the absence of an effect on length changes has also been reported [54]. These discrepancies can be explained by differences in recording techniques, heart-muscle contraction modes, types of heart-muscle preparations, and animal species. We do not observe significant shifts in AP duration with changes in length of guinea pig atrial strips and ventricular trabeculae in steady-state isometric mode (Figure 5B). Our results are in good agreement with a number of data [36,54].

### 3.4. Kinetics of the Ca^2+^ Transient

It has been shown that, with the increasing length of guinea pig atrial and ventricular preparations, the initial phase of the CaT decay is accelerated, and the final phase of the CaT decay is slowed (Figure 6A,C). Previously, we had noted a similar effect in isolated rat preparations, where the final phase was termed the “bump” [34,55]. The non-monotonic CaT decay reflects the complex kinetics of calcium ion sequestration in cardiomyocytes. This can be explained as follows: (1) the rapid CaT decay at the beginning reflects a decrease in [Ca^2+^]_i_ as a result of the joint activity of the calcium sequestration systems (SERCA, Ca-ATP, Na^+^/Ca^2+^ exchanger at the surface membrane and binding to the intracellular buffer system [24,56]); (2) a decrease in [Ca^2+^]_i_ in the myoplasm near the muscle proteins leads to a shift of the Ca-TnC system towards dissociation and consequently to an additional amount of calcium ions at the end of the CaT decay, which is proportional to the degree of overlap of the contractile proteins.

It is noteworthy that in our experiments there is no length-dependent modulation of the duration of the AP (Figure 5B) against a background of an increase in the duration of the CaT (Figure 6B), with stretching of RA strips and RV trabeculae of guinea pigs over the entire length-range studied. This suggests that, firstly, with increasing muscle length there is no change (or a balanced change) in ionic currents across the membrane, including calcium ion influx. Secondly, it suggests that the increase in the duration of the CaT and the “crossover” phenomenon is based on the mechanism of the length-dependent activation of contractile proteins as a result of a cooperative increase in the affinity of calcium ions for TnC proteins. In this case, the initial phase of CaT decay (the II phase of the CaT difference curve) is mainly determined by the action of SERCA and other calcium-sequestration systems that remove calcium ions from the myoplasm. At the end of the CaT decay (the III phase of the CaT difference curve), calcium ions are released in proportion to the degree of elongation of the myocardium in length after dissociation of the Ca-TnC complexes, which act as calcium buffers. The method of CaT difference curves was used to quantify this modulation (Figure 7), see section “4. Materials and methods” and article [55]. The amplitude and relative area under the CaT difference curve in the III phase increase with increasing cardiac muscle stretch for both RA strips and RV trabeculae of guinea pigs, with no significant differences between RA and RV. However, such differences are found in the data we obtained in rats [34].

### 3.5. Summary

In the context of the work presented, it should be emphasised that there is a lack of data in the literature on the effect of the degree of myocardial stretch of different heart chambers on electromechanical coupling, with the exception of our work on rat hearts [34].

We agree with the conclusion of the authors of [36] that the absence of a change in the shape of the AP during stretch does not contribute to the manifestation of the Frank–Starling law.

Changes in the Na^+^/Ca^2+^ exchanger, whose current is responsible for the increased CaTs [57,58] and the increased affinity of TnC for calcium ions [59], are considered to be the mechanism of modulation of the AP shape under the influence of stretching, which is consistent with our results.

## 4. Materials and Methods

### 4.1. Ethical Approval

The animals were cared for according to the Directive 2010/63/EU of the European Parliament and the Guide for the Care and Use of Laboratory Animals published by the US National Institutes of Health (NIH Publication No. 85-23, revised 1985), and their use was approved by the local Institutional Ethics Committee (Protocol No. 14/20, 8 December 2020).

### 4.2. Animals

Experiments were performed on healthy 9-week-old outbred male mongrel guinea pigs (*n* = 21) kept in the vivarium of the Institute under standard conditions (12 h of daylight, free access to water and food).

Fifteen minutes before sacrifice, the guinea pigs were injected intramuscularly with heparin (5000 IU/kg) and the muscle relaxant xylazine (1.0 mL/kg). The animals were removed from the experiment under anaesthesia with isoflurane (3–4% mixed with air).

### 4.3. Histological Investigation of the Walls of the Right Atrium and Ventricle

Right heart samples were fixed in 10% buffered formalin for 48 h. The material was histologically processed using a Leica TP1020 (Leica Biosystems Nussloch GmbH, Nußloch, Germany) automatic tissue processor according to the standard technique. Organ samples were embedded in paraffin using a Leica EG1160 (Leica Biosystems Nussloch GmbH, Germany) embedding station. Histological sections with a thickness of 3–5 μm were prepared using a Leica SM2000R (Leica Biosystems Nussloch GmbH, Germany) sled microtome. Histological sections were prepared according to standard methods and subsequently stained with haematoxylin and eosin or picrosirius red. Tissue sections were analysed using a Leica DM2500 (Leica Microsystems GmbH, Germany) microscope connected to a Leica DFC420 (Leica Microsystems Ltd., Germany) video camera and a personal computer.

### 4.4. Heart Muscle Preparations

The heart was washed in situ with a plegic modified Krebs–Henseleit (K-H) solution with a reduced Ca^2+^ concentration (CaCl2 0.2 mM/L) and the addition of 2,3-butadione monoxime (BDM 30 mM/L) to remove blood. The heart was then removed from the chest cavity and placed in a dissection bath. Right ventricular trabecula or right atrial wall strip (100–500 µm wide, 1–3 mm long) were dissected from the heart and attached to the force transducer and length servomotor rods in an experimental bath. The experimental baths were perfused with modified K-H solution (in mM/L): NaCl 118, KCl 4.7, MgSO_4_ 1.2, KH_2_PO_4_ 1.2, NaHCO_3_ 25, CaCl_2_ 1.2, glucose 11.1, insulin 5 IU, pH adjusted to 7.35 at 25 °C and bubbling with a gas mixture of 95% O_2_ and 5% CO_2_. The accuracy of the measurement of the isometric contraction force was ± 0.02 mN, the accuracy of the length control was ± 1 µm.

### 4.5. Simultaneous Investigation of the Mechanical and Electrical Activity of Heart Muscle Preparations

The AP signals were registered with floating glass microelectrodes (10–15 MΩ) using an Intracellular Electrometer IE-210 (Warner Instrument Corporation, Holliston, MA, USA) system. Borosilicate micropipette blanks were prepared using a KOPF needle/pipette Puller model 730 (David KOPF Instruments, Tujunga, CA, USA). The micropipette blanks were filled with a KCl solution (3 M) using a vacuum pump and a water bath. The microelectrode was connected to an external matching unit of the measuring system using a thin silver chloride wire. The external matching unit of the measuring system was mounted on a micromanipulator (NARISHIGE, model MM-3, Tokyo, Japan). APs were recorded at five lengths (0.8, 0.85, 0.9, 0.95, 1.0 *L_max_*) from isolated muscle preparations of the right atrium and ventricle. The measuring and setting devices of the two-channel setup were connected to a personal computer via ADC/DAC E-502 (L-CARD, Moscow, Russia) to record signals and control servomotors with a frequency of 10 kHz. For online measurements, we used our own software package. Data were obtained at 30 °C and a stimulation frequency of 2 Hz.

### 4.6. Simultaneous Investigation of the Mechanical Activity and Ca^2+^ Transients of the Heart Muscle Preparations

The heart-muscle force signal and the fluorescence signal of Fura-2 AM were recorded simultaneously in a Muscle Research System (Scientific Instruments, Heidelberg, Germany) via a ADC/DAC L-502 (L-CARD, Russia) with a sampling rate of 10 kHz, under the control of our own software package for online measurements. To measure the CaT signal, muscles were incubated in a solution containing 4 µM/L Fura-2 AM + 0.08% *w*/*v* Pluronic F-127. The fluorophore (Fura-2 AM) was excited with light at two frequencies with wavelengths of 340/380 nm, and the dye was emitted at a wavelength of 510 nm. Data were obtained from RA strips (*n* = 6) and RV trabeculae (*n* = 11) at a temperature of 30 °C and a stimulation frequency of 2 Hz.

### 4.7. Force-Length Protocol

The cardiac muscle preparations were adapted to the experimental conditions for 30–45 min before the start of the experiment. The muscles were then shortened to a length at which they developed the lowest amplitude of active tension. Then, the length of the muscle was gradually increased. At each step, the parameters of the contraction force were stabilised. The gradual increase in length was continued until the maximum active tension was reached; this length was taken as maximal—*L_max_*. Then, the experimental protocol was carried out. Measurements of mechanical activity and CaTs or APs were made at lengths of 0.7 to 1.0 *L_max_* for RA strips and 0.8 to 1.0 *L_max_* for RV trabeculae with a discrete 0.05 *L_max_*.

Isometric force signals obtained under steady-state conditions at different lengths were used to obtain the “active tension—length” (or “force-length”) relationship. Mechanical tension was estimated by normalising the force values to the cross-sectional area of the muscle preparation. The muscle cross-sectional area was calculated as S = π×a×b/4, where a and b are the large and small diameters of the ellipse. The values for maximum force development and relaxation rates of isometric tension were normalised to the active tension amplitude to avoid the influence of the amplitude scale factor.

Before analysing the CaT signals at different muscle lengths (or preloads), the autofluorescence signal of the muscle obtained at the same length without Fura-2 AM (before loading the muscle with the dye) was subtracted from the Fura-2 AM luminescence signal. For comparison of temporal characteristics, the luminescence signal corresponding to the diastolic level was subtracted from each CaT signal. The CaT signals obtained in the previous step were normalised by their own amplitude. Thus, the values of the luminescence signals ranged from 0 to 1. The trajectories of the CaT signals were normalised by their amplitude because the amplitude of the CaTs depends on several uncontrolled variables (fading, degree of dye saturation of the cardiomyocytes and variable luminescence area of the myocardial preparation) in addition to calcium ion concentration [60]. “Difference curves” were obtained from normalised CaT signal traces. For this purpose, the CaT signal values obtained at a muscle preparation length of 0.8 *L_max_* were subtracted from the CaT signal at each length after muscle stretch. Three phases were identified in the “CaT difference curves”: the rise phase of the CaT—the I phase; the initial phase of the CaT decay—the II phase; and the late phase of the CaT decay—the III phase. The values of the amplitude and area under the difference curve in the III phase were used to evaluate the effects of the degree of myofilament overlap on Ca^2+^ release from Ca-TnC complexes [55].

### 4.8. Statistical Analysis

The statistical analysis was carried out using the Python programming language, the Scipy.stats, scikit_posthocs, pingouin modules. The following tests were used for data processing: Shapiro–Wilk test for normality of distribution; Friedman test ANOVA, Siegel and Castellan’s all-pairs comparisons post hoc closed method test based on Simes tests (non-negative) to adjust *p* values for multiple comparisons for dependent samples; Mann–Whitney pairwise comparison for independent samples; Spearman rank correlation coefficient, for correlation analysis.

## 5. Conclusions

Morphometric studies show that guinea pig right ventricular cardiomyocytes have a significantly larger diameter than right atrial cardiomyocytes.

During steady-state isometric contractions of the atrial strips and ventricular trabeculae of a guinea pig, the following is observed:-The durations of the isometric contraction, transmembrane action potential, and a^2+^ transient are shorter in the atrial myocardium than in the ventricular myocardium at all length ranges studied;-The amplitude and duration of isometric contraction and the Ca^2+^ transient of atrial and ventricular myocardial preparations increase monotonically with increasing length;-The passive tension of the right atrial strips exceeds the passive tension of the right ventricular trabeculae over the entire length range studied;-The values of the time to peak tension, and the relaxation time from peak to decline to half the tension amplitude, are significantly lower for the right atrial strips compared with those of the right ventricular trabeculae over the entire length range studied;-The normalised values of the maximal rates of isometric tension development and relaxation decrease with the increasing length of the atrial and ventricular myocardial preparations;-The contribution of action potential duration to the heterometric regulation of guinea pig right atrial and ventricular isometric tension is absent;-For the first time, it has been shown that the amplitude and area of the “difference curves” of the Ca^2+^ transients increase with increasing length of both the right atrial strips and ventricular trabeculae of the guinea pig, but the differences in these values between right atrial and ventricular myocardial preparations are statistically insignificant.

It can be assumed that, firstly, the increase in the amplitude of the isometric tension is not related to the duration of the action potential and is determined by the degree of overlap of the contractile proteins in the myocardium of the right atrium and ventricle of the guinea pig. This means that the sensitivity of the heterometric regulation mechanisms is approximately the same. Second, changes in the degree of overlap of the contractile proteins of the guinea pig atrial and ventricular myocardium mainly affect the intracellular Ca^2+^ kinetics and the volume of the calcium pool of TnC activation.

## Figures and Tables

**Figure 1 ijms-24-15524-f001:**
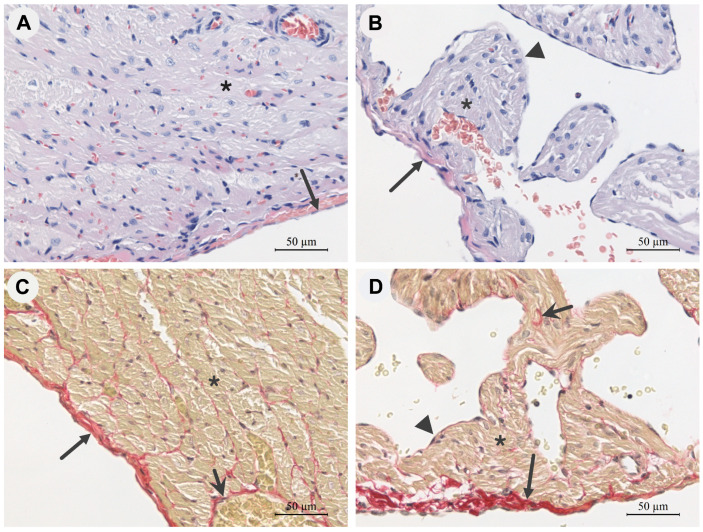
Representative photomicrographs of longitudinal histological sections of the free wall of the right ventricle (**A**,**C**) and the wall of the right atrium (**B**,**D**) of guinea pig hearts under physiological conditions, objective magnification 40×. Haematoxylin and eosin staining (**A**,**B**): cell nuclei—dark purple; cytoplasm and intercellular substance—pink. Picrosirius red staining (**C**,**D**): connective tissue collagen skeleton—dark maroon fibres; cardiomyocytes—yellow-brown; cell nuclei—black. Long arrow—epicardium, short arrow—connective tissue carcass, ▲—endocardium, *—cardiomyocytes.

**Figure 2 ijms-24-15524-f002:**
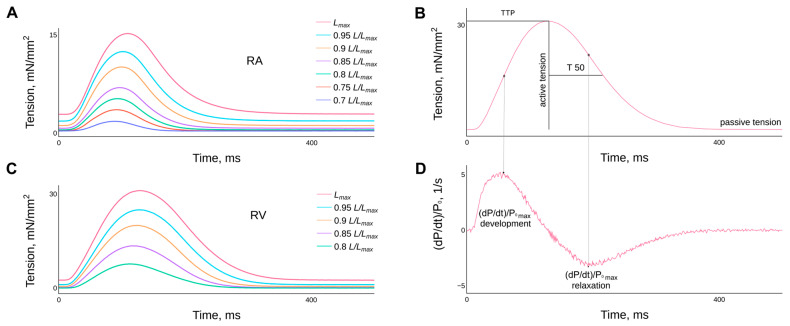
Representative examples of the effect of length (or preload) on the time course of isometric tension in single contraction preparations of right atrial (**A**) and right ventricular (**C**) guinea pig myocardium. The lengths are presented as fractions of *L_max_* and are given in the legend. Panels (**B**,**D**) show the methodology used to determine the characteristics of isometric contraction, such as: the time to peak isometric tension (TTP); the time of tension relaxation from TTP to half the amplitude of active tension in the decay phase (T50); the normalised values of the maximum rates of tension development and relaxation. Temperature 30 °C, stimulation frequency 2 Hz.

**Figure 3 ijms-24-15524-f003:**
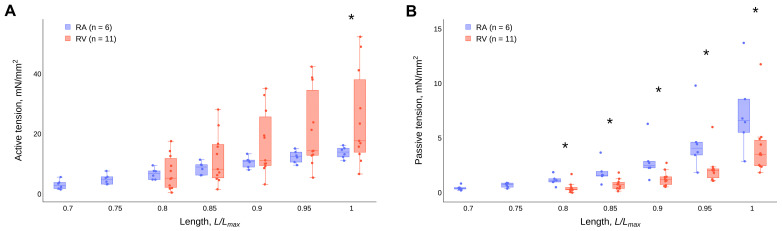
Effect of the length of guinea pig atrial strips and ventricular trabeculae on the magnitude of active isometric tension (**A**) and passive isometric tension (**B**) under steady-state conditions. Data are presented as box-whisker plots (box—IQR (Q1–Q3), Q2 inside the box, whisker—min and max, points—values (scatter plot)). *—statistically significant differences between RA and RV (*p* < 0.05, Mann–Whitney U-test) at the corresponding length.

**Figure 4 ijms-24-15524-f004:**
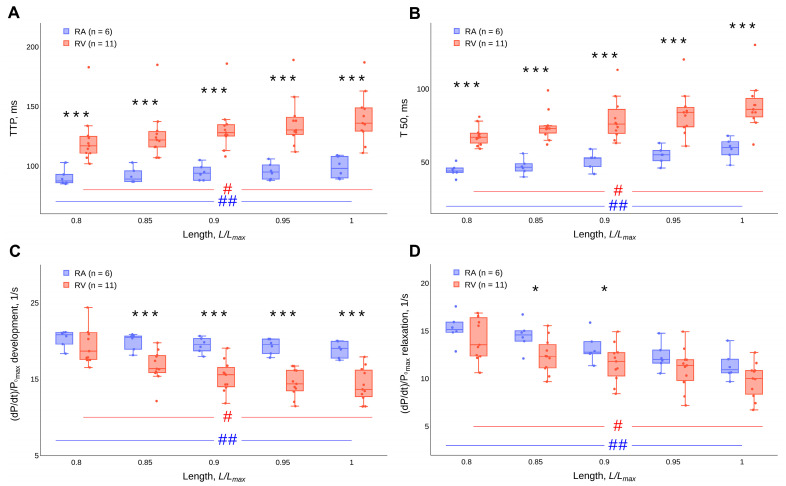
Effect of the length of guinea pig atrial strips and ventricular trabeculae on (**A**) time to peak isometric tension (TTP), (**B**) time of tension relaxation from TTP to half the amplitude of active tension in the relaxation phase (T50), (**C**) the normalised values of maximal rates of tension development and (**D**) relaxation. Data are presented as box-whisker plots (box—IQR (Q1–Q3), Q2 inside the box, whisker—min and max, points—values (scatter plot)). * (*p* < 0.05) and *** (*p* < 0.001)—significant differences between RA and RV at the corresponding length (Mann–Whitney U-test). # and ##—significant differences between different lengths for RV and RA, respectively (*p* < 0.05, Friedman’s ANOVA).

**Figure 5 ijms-24-15524-f005:**
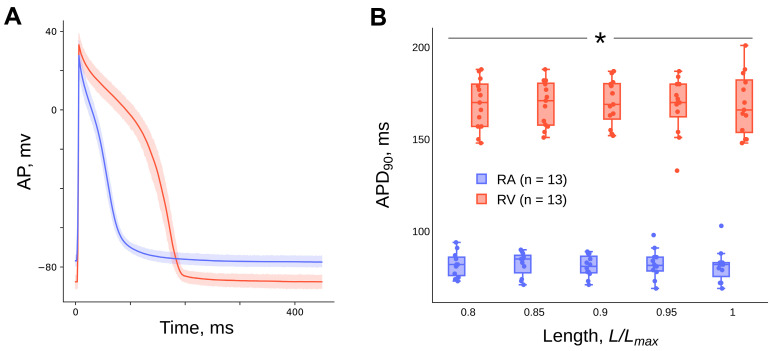
Superposition of the transmembrane action potential trajectories of isolated guinea pig right atrial strips (*n* = 12) and right ventricular trabeculae (*n* = 13) (**A**) at a length of 0.95 *L_max_*. Dependence of action potential duration at the level of amplitude repolarisation by 90% (APD90) of guinea pig right atrial strips and ventricular trabeculae on length (**B**). Data are presented as (**A**) mean ± 95% ci (confidence interval), (**B**) box-whisker plots (box—IQR (Q1–Q3), Q2 inside the box, whisker—min and max, points—values (scatter plot)). Temperature 35 °C, stimulation frequency 2 Hz. *—significant differences between RA and RV at the corresponding length (*p* < 0.05, Mann–Whitney U-test).

**Figure 6 ijms-24-15524-f006:**
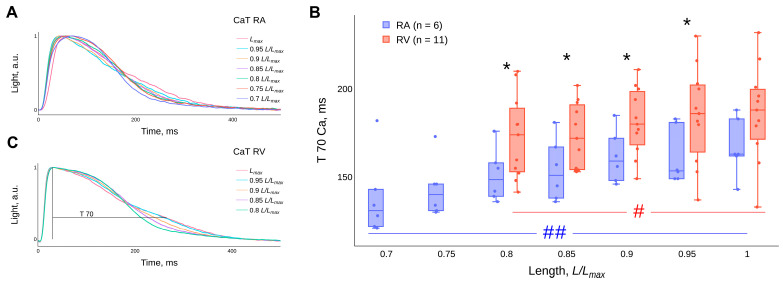
Representative examples of the effect of length (or preload) on the time course of CaTs in single contraction preparations of right atrial (**A**) and right ventricular (**C**) guinea pig myocardium. The lengths are presented as fractions of *L_max_* and are given in the legend. The influence of the length of the RA strips and RV trabeculae on the duration of the decay of CaTs from the peak to the level of the amplitude decay by 70% (T70Ca) (**B**). Temperature 30 °C, stimulation frequency 2 Hz. Data are presented as box-whisker plots (box—IQR (Q1–Q3), Q2 inside the box, whisker—min and max, points—values (scatter plot)). *—significant differences between RA and RV at the corresponding length (*p* < 0.05, Mann–Whitney U-test). # and ##—significant differences between different lengths for RV and RA, respectively (*p* < 0.05, Friedman’s ANOVA).

**Figure 7 ijms-24-15524-f007:**
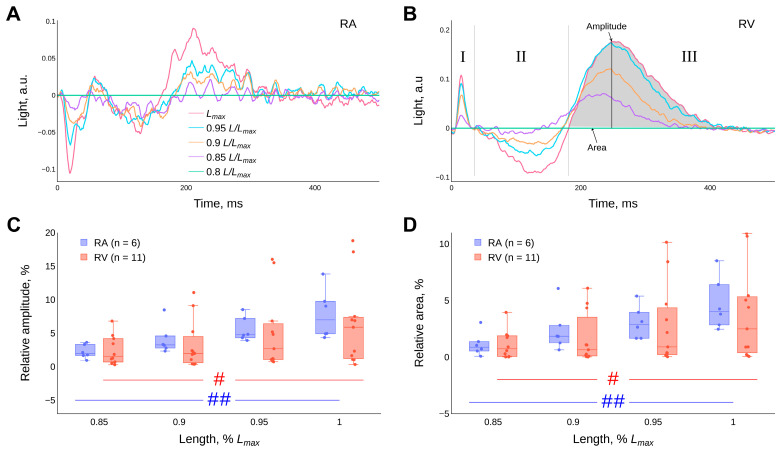
Representative examples of “CaT difference curves” in isolated (**A**) right atrial (RA) strips and (**B**) right ventricular (RV) trabeculae of the guinea pig at different lengths. I, II and III in panel (**B**) indicate the phases of the “CaT difference curves”. Diagrams of the effect of length on the average value of the amplitude of the difference curve in the III phase of the “CaT difference curve” (**C**) and the relative area (in %) calculated under the difference curve in the III phase of the CaT difference curve (**D**). Data are presented as box-whisker plots (box—IQR (Q1–Q3), Q2 inside the box, whisker—min and max, points—values (scatter plot)), panels (**C**,**D**). Temperature 30 °C, stimulation frequency 2 Hz. Significant differences between RA and RV at the corresponding length were not found (*p* < 0.05, Mann–Whitney U-test). # and ##—significant differences between different lengths for RV and RA, respectively (*p* < 0.05, Friedman’s ANOVA).

**Table 1 ijms-24-15524-t001:** Morphometric parameters of the right heart of guinea pig under physiological conditions.

Parameters	Right Atrium (*n* = 8)	Right Ventricle (*n* = 8)	Z	*p*-Value
Wall thickness, μm	26.7 * (3.6, 11.1)	625.2 (46.3, 142.7)	−3.3	0.0002
Cardiomyocyte diameter, μm	7.3 * (0.68, 3.3)	12.7 (1.3, 6.0)	−2.51	0.004
Number of cardiomyocyte nuclei in 1 mm^2^	2461 (469.5, 3090.2)	1637 (286.1, 1883.1)	−1.59	0.06
Myocardial collagen content, %	8.9 * (0.9, 2.9)	6.5 (1.3, 3.9)	−2.36	0.007

All data are presented as mean and 95% confidence intervals. *—indicates statistically significant differences (*p* < 0.05, Mann–Whitney U test) between right ventricular and atrial parameters.

**Table 2 ijms-24-15524-t002:** Spearman correlation coefficients of the normalised values of the maximal rate of force development with the normalised values of the maximal rate of force relaxation of isolated right atrial strips and ventricular trabeculae depending on the strip length.

Length, *L/L_max_*	Right Atrial Strips (*n* = 6)			Right Ventricular Trabeculae (*n* = 11)		
	r	*p*-Value	95% ci	r	*p*-Value	95% ci
0.7	0.49	0.4	(−0.54, 0.93)	—		
0.75	0.71	0.25	(−0.23, 0.97)	—		
0.8	0.94	0.17	(0.56, 0.99)	0.53	0.21	(−0.21, 0.88)
0.85	0.94	0.17	(0.56, 0.99)	0.45	0.27	(−0.3, 0.86)
0.9	0.77	0.23	(−0.11, 0.97)	0.31	0.44	(−0.45, 0.81)
0.95	0.77	0.23	(−0.11, 0.97)	0.37	0.36	(−0.39, 0.83)
1	0.77	0.23	(−0.11, 0.97)	0.32	0.43	(−0.44, 0.81)

## Data Availability

Data generated or analysed during this study are available from the corresponding author upon reasonable request.

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
