# Peer review of "Differences in Mechanical, Electrical and Calcium Transient Performance of the Isolated Right Atrial and Ventricular Myocardium of Guinea Pigs at Different Preloads (Lengths)"

_ijms, 2023, doi:10.3390/ijms242115524_

Round 1

Reviewer 1 Report

This is a very interesting topyc, but you need to improve it:

1. Add an etthical section, you need the approval of ethic cominteee to animal researrch or to manga biological samples, provide the ethic approval number and name and explain that you follow the Kelsinki statemetn.

2. Provide a methods section, you have explain where you oobtained the biological samples, the procedures to manage them and transform in microscopic samples, and the condition of original animals (genetic manipulation, helthy...?). In adition provide a statistic section analysis (software and statistical test used).

3. in results in table 1 provide Z statistic, p values and average difference (95%CI). In table 2, if the variables where non-normal, Spearman correlation is most appropiate, provide 95$CI p values

4. In text I read you perform also a Friedamn test, provide a complete table with all results of this test (mean and sd for each group and measuremetn time, time effect p value, group effect p valuem and group:time p value with their X2 and degrees of freedom and post hoc tests with bonferroni correction  table results, and then comment them in text

Reviewer 2 Report

Thank you for the opportunity to review this manuscript. The authors have examined the physiology of the impact of stretch (sarcomere length) on guinea pig cardiac atria and ventricle, to further understand the mechanism of the Frank-Starling Mechanism. They simultaneously examined the morphometry, force generation, action potential and calcium transients within isolated cardiac tissue. They found that with increasing length, both the isometric contractile amplitude & duration, and the Ca transient (plus uniquely the Ca transient difference curves) increased. However, the rates of the tension development and relaxation decreased. These effects are shown by the authors to be independent of the action potential (a point that has been controversial in the past), consistent with the capacity of the contractile proteins to buffer Ca as a function of length/overlap being primarily responsible. These data add to our understanding of the the Frank-Starling mechanism.

Overall, the paper is well written and clear. I have only a few minor comments that the authors may wish to address:

1. Introduction - the authors published some preliminary data last year that they obtained in rats, to which they refer on a number of occasions. However, for the current work they chose guinea pigs. Perhaps they could provide from background for the rationale for changing species for these experiments? Additionally, it is unclear to me why they preferred working with the right side of the heart rather than the left? Has it to do with the myosin or other contractile protein isoforms that are present on the right vs the left side?

2. On page 4 the authors list lymphocytes and neutrophils as connective tissue cells.  These cells are inflammatory or immune cells that are present adjacent to connective tissue. 

Round 2

Reviewer 1 Report

The paper have been improved, I only hve a last suggesttion, translate section 4 material and methods before resutls, it seens cleraly